# ROBUST EXPLANATION CONSTRAINTS
# FOR NEURAL NETWORKS

**Matthew Wicker**[*,1], **Juyeon Heo**[*,2], **Luca Costabello**[3], **Adrian Weller**[1,2]

[1] The Alan Turing Institute,   [2] University of Cambridge,   [3] Accenture Labs

## ABSTRACT

Post-hoc explanation methods are used with the intent of providing insights about neural networks and are sometimes said to help engender trust in their outputs. However, popular explanations methods have been found to be fragile to minor perturbations of input features or model parameters. Relying on constraint relaxation techniques from non-convex optimization, we develop a method that upper-bounds the largest change an adversary can make to a gradient-based explanation via bounded manipulation of either the input features or model parameters. By propagating a compact input or parameter set as symbolic intervals through the forwards and backwards computations of the neural network we can formally certify the robustness of gradient-based explanations. Our bounds are differentiable, hence we can incorporate provable explanation robustness into neural network training. Empirically, our method surpasses the robustness provided by previous heuristic approaches. We find that our training method is the only method able to learn neural networks with certificates of explanation robustness across all six datasets tested.

## 1  INTRODUCTION

Providing explanations for automated decisions is a principal way to establish trust in machine learning models. In addition to engendering trust with end-users, explanations that reliably highlight key input features provide important information to machine learning engineers who may use them to aid in model debugging and monitoring (Pinto et al., 2019; Adebayo et al., 2020; Bhatt et al., 2020a;b). The importance of explanations has led to regulators considering them as a potential requirement for deployed decision-making algorithms (Gunning, 2016; Goodman & Flaxman, 2017). Unfortunately, deep learning models can return significantly different explanations for nearly identical inputs (Dombrowski et al., 2019) which erodes user trust in the underlying model and leads to a pessimistic outlook on the potential of explainability for neural networks (Rudin, 2019). Developing models that provide robust explanations, i.e., that provide similar explanations for similar inputs, is vital to ensuring that explainability methods have beneficial insights (Lipton, 2018).

Current works evaluate the robustness of explanations in an adversarial setting by finding minor manipulations to the deep learning pipeline which cause the worst-case (e.g., largest) changes to the explanation (Dombrowski et al., 2019; Heo et al., 2019). It has been shown that imperceptible changes to an input can fool explanation methods into placing importance on arbitrary features (Dombrowski et al., 2019; Ghorbani et al., 2019) and that model parameter can be manipulated to globally corrode which features are highlighted by explanation methods (Heo et al., 2019). While practices for improving explanation robustness focus on heuristic methods to avoid current attacks (Dombrowski et al., 2022; Wang et al., 2020; Chen et al., 2019), it is well-known that adversaries can develop more sophisticated attacks to evade these robustness measures (Athalye et al., 2018). To counter this, our work establishes the first framework for general neural networks which provides a guarantee that for any minor perturbation to a given input's features or to the model's parameters, the change in the explanation is bounded. Our bounds are formulated over the input-gradient of the model which is a common source of information for explanations (Sundararajan et al., 2017; Wang et al., 2020). Our guarantees constitute a formal certificate of robustness for a neural network's

---

Author email addresses in listed order: mwicker@turing.ac.uk, jh2324@cam.ac.uk, luca.costabello@accenture.com, aweller@turing.co.uk

explanations at a given input which can provide users, developers, and regulators with a heightened sense of trust. Further, while it is known that explanations of current neural networks are not robust, the differentiable nature of our method allows us to incorporate provable explanation robustness as a constraint at training time, yielding models with significantly heightened explanation robustness.

Formally, our framework abstracts all possible manipulations to an input's features or model's parameters into a hyper-rectangle, a common abstraction in the robustness literature (Mirman et al., 2018; Gowal et al., 2018). We extend known symbolic interval analysis techniques in order to propagate hyper-rectangles through both the forwards and backwards pass operations of a neural network. The result of our method is a hyper-rectangle over the space of explanations that is guaranteed to contain every explanation reachable by an adversary who perturbs features or parameters within the specified bounds. We then provide techniques that prove that all explanations in the reachable explanation set are sufficiently similar and thus that no successful adversarial attack exists. Noticing that smaller reachable explanation sets imply more robust predictions, we introduce a novel regularization scheme which allows us minimize the size of of the explanation set during parameter inference. Analogous to state-of-the-art robustness training (Gowal et al., 2018), this allows users to specify input sets and parameter sets as explainability constraints at train time. Empirically, we test our framework on six datasets of varying complexity from tabular datasets in financial applications to medical imaging datasets. We find that our method outperforms state-of-the-art methods for improving explanation robustness and is the only method that allows for certified explanation robustness even on full-color medical image classification. We highlight the following contributions:

- We instantiate a framework for bounding the largest change to an explanation that is based on the input gradient, therefore certifying that no adversarial explanation exists for a given set of inputs and/or model parameters.

- We compute explicit bounds relying on interval bound propagation, and show that these bounds can be used to regularize neural networks during learning with robust explanations constraints.

- Empirically, our framework allows us to certify robustness of explanations and train networks with robust explanations across six different datasets ranging from financial applications to medical image classification.

## 2 RELATED WORK

Adversarial examples, inputs that have been imperceptibly modified to induce misclassification, are a well-known vulnerability of neural networks (Szegedy et al., 2013; Goodfellow et al., 2015; Madry et al., 2018). A significant amount of research has studied methods for proving that no adversary can change the prediction for a given input and perturbation budget (Tjeng et al., 2017; Weng et al., 2018; Fazlyab et al., 2019). More recently, analogous attacks on gradient-based explanations have been explored (Ghorbani et al., 2019; Dombrowski et al., 2019; Heo et al., 2019). In (Ghorbani et al., 2019; Dombrowski et al., 2019) the authors investigate how to maximally perturb the explanation using first-order methods similar to the projected gradient descent attack on predictions (Madry et al., 2018). In (Heo et al., 2019; Slack et al., 2020; Lakkaraju & Bastani, 2020; Dimanov et al., 2020) the authors investigate perturbing model parameters rather than inputs with the goal of finding a model that globally produces corrupted explanations while maintaining model performance. This attack on explanations has been extended to a worrying use-case of disguising model bias (Dimanov et al., 2020).

Methods that seek to remedy the lack of robustness work by either modifying the training procedure (Dombrowski et al., 2022; Wang et al., 2020) or by modifying the explanation method (Anders et al., 2020; Wang et al., 2020; Si et al., 2021). Methods that modify the model include normalization of the Hessian norm during training in order to give a penalty on principle curvatures (Dombrowski et al., 2022; Wang et al., 2020). Further work suggests attributional adversarial training, searching for the points that maximize the distance between local gradients (Chen et al., 2019; Ivankay et al., 2020; Singh et al., 2020; Lakkaraju et al., 2020). Works that seek to improve explanation in a model agnostic way include smoothing the gradient by adding random noise (Smilkov et al., 2017; Sundararajan et al., 2017) or by using an ensemble of explanation methods (Rieger & Hansen, 2020)

The above defenses cannot rule out potential success of more sophisticated adversaries and it is well-known that the approaches proposed for improving the robustness of explanations do not work

against adaptive adversaries in the case of prediction robustness (Athalye et al., 2018; He et al., 2017). However, *certification* methods have emerged which provide sound guarantees that even the most powerful adversary cannot fool a models prediction (Wicker et al., 2018; Gehr et al., 2018; Wu et al., 2020). Moreover, these methods have been made differentiable and used for training models with provable prediction robustness and for small networks achieve state-of-the-art robustness results (Gowal et al., 2018; Mirman et al., 2018; Wicker et al., 2021). This work extends these certification methods to the problem of robust explanations. By adapting results from symbolic interval analysis of Bayesian neural networks (Wicker et al., 2020; Berrada et al., 2021) and applying them to interval analysis of both the forward pass *and* backwards pass we are able to get guarantees on the robustness of gradient-based explanations. Appendix J summarizes literature on non-gradient-based explanations.

## 3 BACKGROUND

We consider neural networks tasked with solving a supervised learning problem where we are given a dataset of $n_{\mathcal{D}}$-many inputs and labels, $\mathcal{D} = \{x^{(i)}, y^{(i)}\}_{i=0}^{n_{\mathcal{D}}-1}$, with inputs $x^{(i)} \in \mathbb{R}^n$, and corresponding target outputs $y^{(i)} \in \mathbb{R}^m$ either a one-hot class vector for classification or a real-valued vector for regression. We consider a feed forward neural network (NN) as a function $f^\theta : \mathbb{R}^n \to \mathbb{R}^m$, parametrised by a vector $\theta \in \mathbb{R}^p$ containing all the weights and biases of the network. Given a NN $f^\theta$ composed of $K$ layers, we denote by $f^{\theta,1}, ..., f^{\theta,K}$ the layers of $f^\theta$ and we have that the parameters are given by the set, $\theta = \left(\{W^{(i)}\}_{i=1}^K\right) \cup \left(\{b^{(i)}\}_{i=1}^K\right)$, where $W^{(i)}$ and $b^{(i)}$ represent weights and biases of the $i$−th layer of $f^\theta$. Moreover, we take $\sigma$ to be the activation function of the network and $\mathcal{L}$ to be the loss function. Given this, we can write down the forward and backwards pass w.r.t. an input $x$ through a network network as follows:

<table>
<tr><td>Forward Pass:</td><td>Backward Pass:</td></tr>
<tr><td>

$$z^{(0)} = x$$
$$\zeta^{(k)} = W^{(k)} z^{(k-1)} + b^{(k)}$$
$$z^{(k)} = \sigma(\zeta^{(k)})$$

</td><td>

$$d^{(K)} = \partial\mathcal{L}/\partial z^{(K)}$$
$$\delta^{(k-1)} = \frac{\partial z^{(k)}}{\partial \zeta^{(k)}} \frac{\partial \zeta^{(k)}}{\partial z^{(k-1)}}$$
$$d^{(k-1)} = d^{(k)} \odot \delta^{(k-1)}$$

</td></tr>
</table>

where $\odot$ represents the Hadamard or element-wise product. We highlight that $d^{(0)} = \partial\mathcal{L}/\partial x$, the gradient of the loss with respect to the input, and will refer to this with the vector $v$, i.e., $v = \nabla_x \mathcal{L}(f^\theta(x))$. The input gradient tells us how the NN prediction changes locally around the point $x$ and is therefore a primary source of information for many explanation methods (Sundararajan et al., 2017; Zintgraf et al., 2017; Smilkov et al., 2017).

### 3.1 ATTACKS ON EXPLANATIONS

**Input Perturbations** Analogous to the standard adversarial attack threat model, targeted input perturbations that attack explanations seek to modify inputs to achieve a particular explanation. We denote an adversary's desired explanation vector as $v^{\text{targ}} \in \mathbb{R}^n$. The goal of the adversary is to find an input $x^{\text{adv}}$ such that $h(v^{\text{targ}}, v^{\text{adv}}) \leq \tau$ where $v^{\text{adv}} = \nabla_{x^{\text{adv}}} \mathcal{L}(f^\theta(x^{\text{adv}}))$, $h$ is a similarity metric, and $\tau$ is a similarity threshold that determines when the attack is sufficiently close to the target. The structural similarity metric, Pearson correlation coefficient, and mean squared error have all been investigated as forms of $h$ (Adebayo et al., 2018). We use the mean squarred error for $h$ throughout this paper and discuss cosine similarity in Appendix H. Moreover, for the untargeted attack scenario, an attacker simply wants to corrupt the prediction as much as possible. Thus no target vector is set and an attack is successful if $h(v, v^{\text{adv}}) \geq \tau$ where $\tau$ is a threshold of dissimilarity between the original explanation and the adversarially perturbed explanation.

**Model Perturbations** Not typically relevant when attacking predictions is the model parameter perturbation setting. In this setting, an attacker wants to perturb the model in a way that maintains predictive performance while obfuscating any understanding that might be gained through explanations. As before, we assume the adversary has a target explanation $v^{\text{targ}}$ and that the adversary is successful if they find a parameter $\theta^{\text{adv}}$ such that $h(v^{\text{targ}}, v^{\text{adv}}) \leq \tau$ where $v^{\text{adv}} = \nabla_x \mathcal{L}(f^{\theta^{\text{adv}}}(x))$. The untargeted model attack setting follows the same formulation as in the input attack setting.

Model parameter attacks on explanations are usually global and thus one tries to find a single value of $\theta^{\text{adv}}$ that corrupts the model explanation for many inputs at once. This has been investigated in (Heo et al., 2019; Slack et al., 2020; Lakkaraju & Bastani, 2020) and in (Anders et al., 2020; Dimanov et al., 2020) where the authors explore modifying models to conceal discriminatory bias.

## 4 PROBLEM STATEMENTS

In this section, we formulate the fragility (the opposite of robustness) of an explanation as the largest difference between the maximum and minimum values of the input gradient for any input inside of the set $T$ and/or any model inside the set $M$. We then show how one can use the maximum and minimum input gradients to rule out the existence of successful targeted and untargeted attacks. We start by formally stating the key assumptions of our method and then proceed to define definitions for robustness. *Assumptions:* the methods stated in the text assume that input sets and weight sets are represented as intervals, we also assume that all activation functions are monotonic, and that all layers are affine transformations e.g., fully connected layers or convolutional layers. We discuss when and how one can relax these assumptions in Appendix A. Now, we proceed define what it means for a neural network's explanation to be robust around an input $x \in \mathbb{R}^n$.

**Definition 1** *Input-Robust Explanation Given a neural network $f^\theta$, a test input $x$, a loss function $\mathcal{L}$, a compact input set $T$, and a vector $\delta \in \mathbb{R}^n$, we say that the explanation of the network $f$ is $\delta$-input-robust iff $\forall i \in [n]$:*

$$\left\| \min_{x \in T} \frac{\partial \mathcal{L}}{\partial x_i} - \max_{x \in T} \frac{\partial \mathcal{L}}{\partial x_i} \right\| \leq \delta_i \tag{1}$$

*Moreover we define the set $E_T := \{v' \mid \forall i \in [n], \min_{x \in T} \partial \mathcal{L}/\partial x_i \leq v'_i \leq \max_{x \in T} \partial \mathcal{L}/\partial x_i\}$ to be the reachable set of explanations.*

An intuitive interpretation of this definition is as a guarantee that for the given neural network, input, and loss, there does not exist an adversary that can change the input gradient to be outside of the set $E_T$, given that the adversary only perturbs the input by amount specified by $T$. Further, the vector $\delta$ defines the per-feature-dimension fragility (lack of robustness). Under the interval assumption, we take $T$ to be $[x^L, x^U]$ such that $\forall i \in [n], x_i^L \leq x_i \leq x_i^U$. For simplicity, we describe this interval with a width vector $\epsilon \in \mathbb{R}^n$ such that $[x^L, x^U] = [x - \epsilon, x + \epsilon]$. We now present the analogous definition for the model perturbation case.

**Definition 2** *Model-Robust Explanation Given a neural network $f^\theta$, a test input $x$, a loss function $\mathcal{L}$, a compact parameter set $M$, and a vector $\delta \in \mathbb{R}^n$ we say that the explanation of the network $f$ is $\delta$-model-robust iff $\forall i \in [n]$: $\|\min_{\theta \in M} \partial \mathcal{L}/\partial x_i - \max_{\theta \in M} \partial \mathcal{L}/\partial x_i\| \leq \delta_i$. Moreover, we define $E_M := \{v' \mid \forall i \in [n], \min_{\theta \in M} \partial \mathcal{L}/\partial x_i \leq v'_i \leq \max_{\theta \in M} \partial \mathcal{L}/\partial x_i\}$.*

This definition is identical to Definition 1 save for the the fact that here the guarantee is over the model parameters. As such, a model that satisfies this definition guarantees that for the given neural network and input, there does not exist an adversary that can change the explanation outside of $E_M$ given that it only perturbs the parameters by amount specified by $M$. Similar to what is expressed for inputs, we express $M$ as an interval over parameter space parameterized by a width vector $\gamma \in \mathbb{R}^{n_p}$. To avoid issues of scale, we express an interval over a weight vector $W$ w.r.t. $\gamma$ as $[W - \gamma|W|, W + \gamma|W|]$. Before proceeding we remark on the choice of $\mathcal{L}$. Typically, one takes the input-gradient of the predicted class to be the explanation; however, one may want to generate the class-wise explanation for not just the predicted class, but other classes as well (Zhou et al., 2016; Selvaraju et al., 2017). In addition, adversarial robustness literature often considers attacks with respect to surrogate loss functions as valid attack scenarios (Carlini & Wagner, 2017). By keeping $\mathcal{L}$ flexible in our formulation, we model each of these possibilities.

### 4.1 FROM ROBUST EXPLANATIONS TO CERTIFIED EXPLANATIONS

In this section, we describe the procedure for going from $\delta$-robustness (input or model robustness) to certification against targeted or untargeted attacks. For ease of notation we introduce the notation $v^U$ for the maximum gradient ($\max_{x \in T} \nabla_x \mathcal{L}(f^\theta(x))$ or $\max_{\theta \in M} \nabla_x \mathcal{L}(f^\theta(x))$) and $v^L$ for

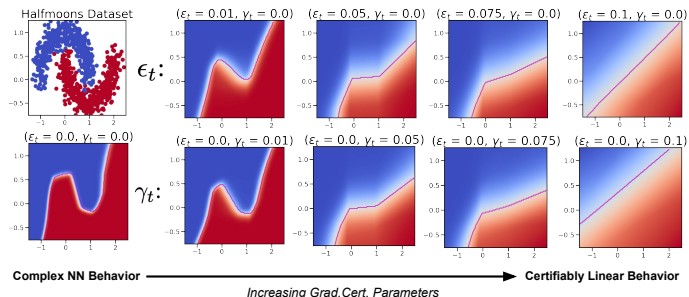

Figure 1: By increasing the parameters of our method, we are able to enforce certifiable locally-linear behavior of an NN model on the half-moons dataset, training data visualized in the top left corner. In the extreme case, illustrated on the far right, we recover a globally linear classifier. The top row varies $\epsilon_t$ the size of the input region while the bottom row varies $\gamma_t$ which enforces locally linear behavior w.r.t. the model parameters.

the minimum gradient ($\min_{x \in T} \nabla_x L(f^\theta(x))$ or $\min_{\theta \in M} \nabla_x \mathcal{L}(f^\theta(x))$). This notation leads to $\delta = |v^U - v^L|$ and $E = [v^L, v^U]$ according to Definitions 1 and 2. Below we discuss how to use these values to certify that no successful targeted or untargeted attack exists.

*Certification of Robustness to Targeted Attacks:* To certify that no successful targeted attack exists it is sufficient to check that $\forall v' \in E, h(v', v^{\text{targ}}) > \tau$. In order to do so, we can define $v^{\text{cert}} := \arg\min_{v \in E} h(v, v^{\text{targ}})$. If the inequality $h(v^{\text{cert}}, v^{\text{targ}}) > \tau$ holds, then we have proven that no successful adversary exists. Where $h$ is the mean squared error similarity, we can construct $v^{\text{cert}}$ from the set $E$ that maximizes the similarity by taking each component $v_i^{\text{cert}}$ to be exactly equal to $v_i^{\text{targ}}$ when $v_i^{\text{targ}} \in [v_i^L, v_i^U]$, we take $v_i^{\text{cert}} = v_i^U$ when $v_i^{\text{targ}} > v_i^U$ and we take $v_i^{\text{cert}} = v_i^L$ otherwise. This construction follows from the fact that minimizing the mean squared error between the two vectors is accomplished by minimizing the mean squared error for each dimension.

*Certification of Robustness to Untargeted Attacks:* Similar to the argument above, we can certify the robustness to untargeted attacks by showing that $\forall v' \in E, h(v, v') \leq \tau$ where $v$ is the original input gradient. As before, if the $v^{\text{cert}} := \arg\max_{v' \in E} h(v, v')$ has mean squared error less than $\tau$ then we have certified that no successful untargeted attack exists. We construct $v^{\text{cert}}$ in this case by taking each component $v_i^{\text{cert}} = v_i^U$ when $|v_i^U - v_i| > |v_i - v_i^L|$ and $v_i^{\text{cert}} = v_i^L$ otherwise.

## 5 COMPUTATIONAL FRAMEWORK

In this section we will provide a computationally efficient framework using interval bound propagation (IBP) to compute values for $v^L$ and $v^U$. We highlight that the values we compute are *sound* but not *complete*. In short, this means the values for $v^L$ computed here are not the largest values for the minimum that can be obtained and the values for $v^U$ are not the smallest values for the maximum that can be obtained. Tighter bounds may be achieved at the cost of greater computational complexity as discussed in Appendix A; however the method we have chosen has computational cost equivalent to four forward passes through the network and is therefore very fast. In order to compute sound $v^L$ and $v^U$ we propagate input and/or model intervals through the forward and backwards pass of the neural network; however, standard propagation techniques do not accommodate joint propagation of input and weight intervals which is necessary for the backwards pass. To solve this problem, we present Lemma 1:

**Lemma 1** (*Wicker, 2021*) *Given an interval over matrices $A^L, A^U \in \mathbb{R}^{a \times b}$ s.t. $\forall i, j$ it holds that $A_{i,j}^L \leq A_{i,j}^U$, and an interval over matrices $B^L, B^U \in \mathbb{R}^{b \times c}$ s.t. $\forall i, j \ B_{i,j}^L \leq B_{i,j}^U$. We denote the center and width of a matrix interval as $B^\mu = (B^L + B^U)/2$ and $B^r = (B^U - B^L)/2$. We also denote $M^B = |A|B^r$ and correspondingly, $M^A = A^r|B|$. We then have the following element-wise inequalities $\forall A^* \in [A^L, A^U]$ and $\forall B^* \in [B^L, B^U]$:*

$$\forall i, j \quad A^\mu B_{i,j}^\mu - M_{i,j}^B - M_{i,j}^A - Q_{i,j} \leq A^* B_{i,j}^* \leq A^\mu B_{i,j}^\mu + M_{i,j}^B + M_{i,j}^A + Q_{i,j}$$

*where $|\cdot|$ indicates element-wise absolute value and $Q = |A^r||B^r|$. We denote a function that returns these matrix multiplication upper and lower bounds given the four parameters above as: $\mathcal{L}(A^L, A^U, B^L, B^U)$ for the lower bound and $\mathcal{U}(A^L, A^U, B^L, B^U)$ for the upper bound.*

As desired, Lemma 1 allows us to jointly propagate intervals over inputs and parameters through affine transformations. We can now restate the forwards and backwards pass in terms of lower and upper bounds on input features and model parameters:

Forward Pass w.r.t. Bounds:

$$z^{(L,0)} = x^L \quad z^{(U,0)} = x^U$$

$$\zeta^{(L,k+1)} = \mathcal{L}(W^{(L,k)}, W^{(U,k)}, z^{(L,k)}, z^{(U,k)})$$

$$\zeta^{(U,k+1)} = \mathcal{U}(W^{(L,k)}, W^{(U,k)}, z^{(L,k)}, z^{(U,k)})$$

$$z^{(L,k)} = \sigma(\zeta^{(L,k)} + b^{(L,k)})$$

$$z^{(U,k)} = \sigma(\zeta^{(U,k)} + b^{(U,k)})$$

Backward Pass w.r.t. Bounds:

$$d^{(L,K)} = \partial L / \partial z^{(L,K)} \quad d^{(U,K)} = \partial L / \partial z^{(U,K)}$$

$$\delta^{(L,k-1)} = \mathcal{L}\left(\frac{\partial z^{(L,k)}}{\partial \zeta^{(k)}}, \frac{\partial z^{(U,k)}}{\partial \zeta^{(k)}}, \frac{\partial \zeta^{(L,k)}}{\partial z^{(k-1)}}, \frac{\partial \zeta^{(U,k)}}{\partial z^{(k-1)}}\right)$$

$$\delta^{(U,k-1)} = \mathcal{U}\left(\frac{\partial z^{(L,k)}}{\partial \zeta^{(k)}}, \frac{\partial z^{(U,k)}}{\partial \zeta^{(k)}}, \frac{\partial \zeta^{(L,k)}}{\partial z^{(k-1)}}, \frac{\partial \zeta^{(U,k)}}{\partial z^{(k-1)}}\right)$$

$$d^{(L,k-1)} = \text{MaxMul}(d^{(U,k)}, d^{(L,k)}, \delta^{(U,k-1)}, \delta^{(L,k-1)})$$

$$d^{(U,k-1)} = \text{MinMul}(d^{(U,k)}, d^{(L,k)}, \delta^{(U,k-1)}, \delta^{(L,k-1)})$$

where $\text{MaxMul}(A, B, C, D)$ is short-hand for the elementwise maximum of the products $\{AB, AD, BC, BD\}$ and the MinMul is defined analgously for the elementwise minimum. Following the above iterative equations we arrive at $d^{L,0}$ (lower bound) and $d^{U,0}$ (upper bound) which are upper and lower bounds on the input gradient for all inputs in $[x^L, x^U]$ and for all parameters in $[\theta^L, \theta^U]$. Formally, these bounds satisfy that $\forall i \in [n]$, $d_i^{L,0} \leq v_i^L$ and $d_i^{U,0} \geq v_i^U$. We provide more detailed exposition and proof in Appendix I. Finally, we can write $\delta^{\text{cert}} = d^{U,0} - d^{L,0}$ and $E^{\text{cert}} = [d^{L,0}, d^{U,0}]$ which satisfy the conditions of Definition 1 and 2.

## 5.1 GRADIENT CERTIFICATION TRAINING LOSS

It follows from Definitions 1 and 2 that models with smaller $\delta$ values are more robust. To minimize $\delta$ during training, it is natural to add $\sum_{i=0}^{n-1} \delta_i$ as a regularization term to the network loss. At training time, we consider the input perturbation budget $\epsilon_t$ as well as the model perturbation budget $\gamma_t$. The input interval is simply taken to be $[x - \epsilon_t, x + \epsilon_t]$ and the interval over a weight parameter $W$ is $[W - |W|\gamma_t, W + |W|\gamma_t]$. Given such intervals, we can then state a function $D(x, \theta, \epsilon_t, \gamma_t)$ which returns the vector $\sum_{i=0}^{n-1} \delta_i^{\text{cert}}$. Thus, minimizing $D(x, \theta, \epsilon_t, \gamma_t)$ corresponds to minimizing an upper bound on the maximum difference in the first derivative leading to improved certification. The complete, regularized loss which we term the gradient certification (Grad. Cert.) loss is:

$$\mathcal{L}_{\text{Grad. Cert.}} = \mathcal{L}(f^\theta(x)) + \alpha D(x, \theta, \epsilon_t, \gamma_t)$$

where $\alpha$ is a parameter that balances the standard loss and the adversarial loss. Throughout the paper, we will continue to refer to the values of $\epsilon$ and $\gamma$ that are used at training time as $\epsilon_t$ and $\gamma_t$. The optimal loss for our regularization term is achieved for an input point $x$ when $D(x, \theta, \epsilon_t, \gamma_t) = 0$. For input robustness, implies that for all points in set $T$ the gradient with respect to the input is constant. Thus, one perspective on our regularization is that it enforces certifiably locally linear behavior of the neural network for all points in $T$. For $\gamma_t$ it is a similar guarantee save that all of the models in $M$ must have point-wise linear behavior at $x$. In Figure 1 we visualize the effect of taking our regularization term to the extreme in a planar classification problem. We see that indeed this perspective holds and that increasing the parameters $\epsilon_t$ or $\gamma_t$ leads to the classifier having more linear behavior.

## 6 EXPERIMENTS

In this section, we empirically validate our approach to certification of explanation robustness as well as our regularization approach to training NNs with certifiably robust explanations. We start by covering the evaluation metrics we will use throughout this section to benchmark robustness of explanations. We analyze robustness of explanations on two financial benchmarks, the MNIST handwritten digit dataset, and two medical image datasets. We report exact training hyper-parameters and details in the Appendix along with further ablation studies, Appendix B, C, D, E, G, and discussion of assumptions and limitations in Appendix A.

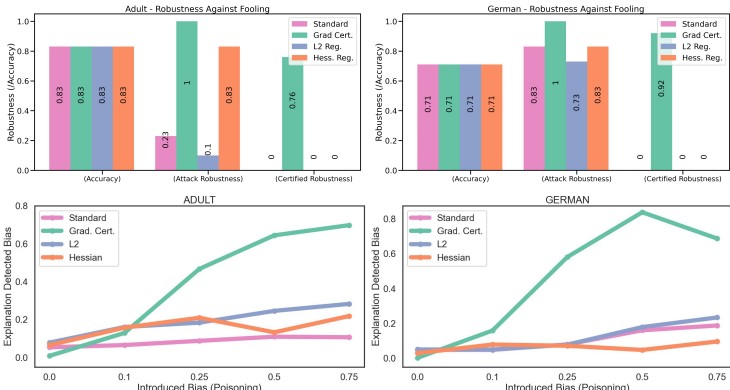

Figure 2: **Top Row:** Only training with our gradient certified loss gives non-trivial certified robustness performance (bars labelled *Certified Robustness*). We find no drop in accuracy when using our method (bars labelled *Accuracy*), and in addition we have attack robustness that surpasses baseline models (bars labelled *Attack Robustness*). **Bottom Row:** We introduce a variable amount of label poisoning (x-axis) to introduce bias into the datasets, and then test to see if the explanation indicates that it is using sensitive features. Looking at the magnitude of the gradients, we observe that gradient certified training forces the model to reveal its sensitivity much more readily.

**Evaluation Metrics** In this section we state the metrics that we will use throughout our analysis. Each of the metrics below are stated for a set of $j$ test inputs $\{x^{(i)}\}_{i=0}^{j-1}$ and for each application we will explicitly state the $\tau$ used.

*Attack robustness:* Given a neural network $f^\theta$, we denote $v^{(i),\text{adv}}$ as the input gradient resulting from an attack method on the test input $x^{(i)}$. We then define the metric as: $1/j \sum_{i=0}^{j-1} \mathbb{1}(h(v^{\text{targ}}, v^{(i),\text{adv}}) > \tau)$. When $v^{(i),\text{adv}}$ is the result of (Dombrowski et al., 2019), a first-order attack in the input space using PGD, we call this *input attack robustness*. Whereas if $v^{(i),\text{adv}}$ is the result of a locally bounded version of the attack in (Heo et al., 2019), a first-order attack in the parameter space using PGD, we call this *model attack robustness*. Intuitively, this measures the proportion of inputs that are robust to various attack methods.

*Certified robustness:* Using our certification method, we compute $v^{(i),\text{cert}}$ according to Section 4.1 and then measure: $1/j \sum_{i=0}^{j-1} \mathbb{1}(h(v^{\text{targ}}, v_i^{(i),\text{cert}}) > \tau)$. Intuitively, this metric is the proportion of inputs which are certified to be robust. Where $v^{(i),\text{cert}}$ is the result of propagating an input set $T$ this is *input certified robustness*, and when propagating $M$ we call it *model certified robustness*.

## 6.1 CERTIFYING EXPLANATIONS IN FINANCIAL DATASETS

In financial applications, it is of interest that users not be able to game financial models into providing incorrect explanations by slightly changing their features. We study the robustness of NNs in financial applications to adversarial users attempting to game the NN into providing an explanation that shows the model is predicting on the basis of protected attributes (e.g., gender, race, age). In order to do so we study the *Adult* dataset, predicting whether or not an individual makes more or less than fifty thousand dollars a year, and the *German Credit* dataset, predicting whether an individual has a good or bad credit rating. Importantly, financial modellers should rigorously ensure/certify that their models are indeed not biased (Benussi et al.); we discuss this further in Appendix D. We assess the robustness against gaming by taking $v^{\text{targ}}$ to be a vector with one entries for each of the sensitive features and zeros elsewhere, and we say an attack is successful if the adversary is able to force the sensitive feature (gender) into being in the top 5 explaining features by magnitude. That is, an attack is successful if the explanation reveals that the sensitive feature is one of the most relied upon features. In the top row of Figure 2 we present the accuracy and robustness values for standard training, gradient certified training (ours), and other robust explanation baselines including those suggested in (Dombrowski et al., 2022), Hessian regularization + softplus activation functions, and (Drucker & Le Cun, 1992), training to minimize $\ell_2$ gradient difference in a ball around the inputs. Objectives, further baselines, and details can be found in Appendix G. On the Adult dataset we find

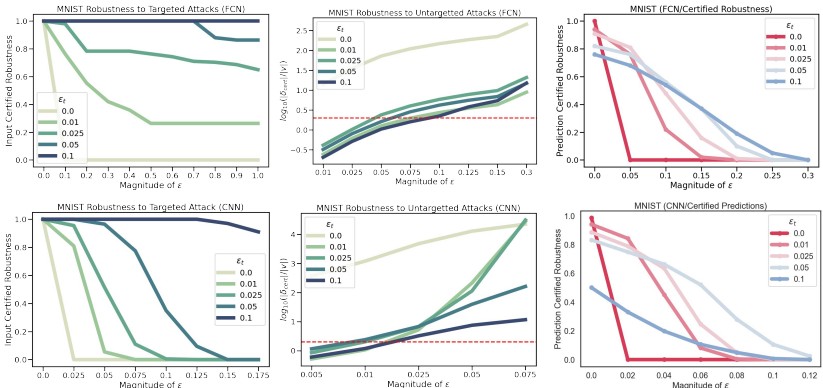

Figure 3: Gradient certified training leads to neural networks with considerably heightened robustness against targeted (left column) and untargeted attacks (center column). In addition, we find that our training leads to certifiably robust predictions (right column). **Left column:** Targeted attack robustness, higher is better, against adversary who tries to force explanation into the corner of an image. We see that FCNs are robust to this attack over the entire domain ($\epsilon = 1.0$). **Center column:** On the y-axis we plot the the average value of $\delta$ normalized by the gradient magnitude, lower is better. We plot a red dashed line at the success threshold for untargeted attacks and find that FCNs are considerably more robust to untargeted attacks. **Right column:** On the y-axis we plot the certified robust accuracy, showing that gradient certified training leads to certified prediction robustness.

that networks trained with (Drucker & Le Cun, 1992) regularization and without any regularization (standard) are vulnerable to explanation attacks. On the German dataset, we find that overall, models are more robust to being fooled. However, we consistently find that the best robustness performance and the only certified behavior comes from gradient certified training.

*Bias detection study* We study the ability of explanations to flag bias in models. We create biased models by performing label poisoning. We set a random proportion $p$ of individuals from the majority class to have a positive classification and the same proportion $p$ of individuals from the minority class to have a negative classification. The proportion $p$ is labeled 'induced bias' in Figure 2 is strongly correlated with the actual bias of the model, see Appendix D. We then use the input gradient, $v$, to indicate if the explanation detects the bias in the learned model. This is done by measuring the proportion $|v_j|/\sum_{i=0}^{n-1}|v_i|$ where $j$ is the index of the sensitive feature this proportion is called 'explanation detected bias' in our plots. We highlight that our method significantly out-performs other methods at detecting bias in the model likely due to the certified local linearity providing more calibrated explanations. While our method is promising for bias detection it should supplement rather than replace a thorough fairness evaluation (Barocas & Selbst, 2016).

## 6.2 CERTIFYING EXPLANATIONS IN DIGIT RECOGNITION

In this section, we study the affect of input peturbations on the explanations for networks trained on the MNIST digit recognition dataset. We train two networks, a two-layer fully-connected neural network with 128 neurons per layer and a four layer convolutional neural network inspired by the architecture in (Gowal et al., 2018).We take a set of twenty target explanations to be binary masks with a five by five grid of ones in the corners with varying distances from the border of the image and based on visual inspection set $\tau = 0.04$, see Figure 8 in the Appendix for illustration. In Figure 3, we plot the result of our robustness analysis. We note that there is a consistent 2% test-set accuracy penalty for every 0.01 increase in $\epsilon_t$. We find that for both FCNs and CNNs we are able to get strong certification performance. We highlight that for FCNs we are able to certify that there is no input in the entire domain that puts its explanation in any corner of the input image, this is indicated by $\epsilon = 1.0$, as the images are normalized in the range $[0, 1]$. In the center column of Figure 3, we plot the results of our untargeted attack analysis. In particular we take the untargeted $v^{\text{cert}}$ described in Section 4.1 and measure $||v^{\text{cert}}||_2/||v||_2$. Again based on visual inspection, we set the untargeted threshold $\tau = 2.0$, indicated with a red dashed line. Finally, in the right-most column of Figure 3,

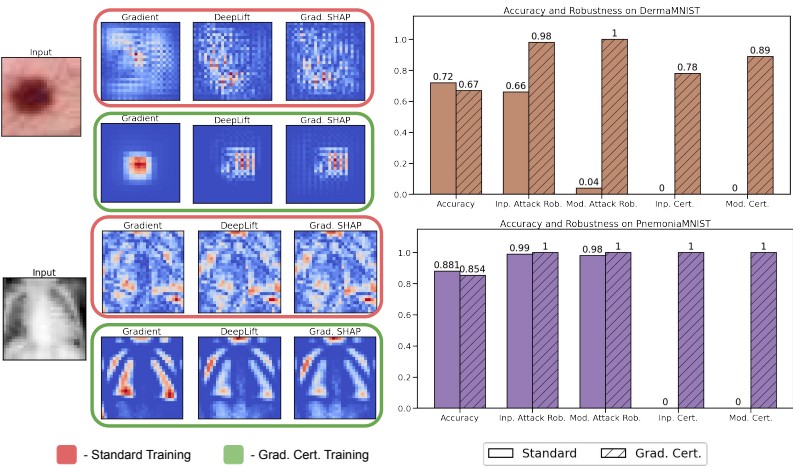

Figure 4: On MedMNIST datasets the gradient certified loss improves attack robustness and gets strong input and model certified robustness. In addition, we find that explanations from a variety of methods are more sparse and intuitive than explanations from normally trained networks. **Top rows:** On DermaMNIST, the gradient certified loss leads to considerable increases in all robustness metrics for only a 5% reduction in test-set accuracy. **Bottom rows:** On PneumoniaMNIST despite standard training having strong attack robustness performance only gradient certified training has non-trival certified robustness with only a 3% reduction in test-set accuracy.

we plot how our training method affects the certifiable prediction robustness. Using bounds from (Gowal et al., 2018) we can certify that no input causes the prediction to change. We show that for both FCNs and CNNs our bounds lead to considerable, non-trivial certified prediction robustness.

## 6.3 CERTIFYING EXPLANATIONS IN MEDICAL IMAGE CLASSIFICATION

We consider two datasets from the MedMNIST benchmark and a third in Appendix E. The first task, *DermaMNIST*, is to classify images into one of 11 different skin conditions. The second task, *PneumoniaMNIST*, is to classify chest x-rays into positive or negative diagnosis for pneumonia. DermaMNIST is represented by full-color images (2352 feature dimensions) while PneumoniaMNIST is represented by black and white images (784 feature dimensions). We use the same 20 binary corner masks as in the MNIST analysis. We train models with $\epsilon_t = 0.01$ and $\gamma_t = 0.01$ and test with $\epsilon = 0.025$ and $\gamma = 0.025$. Across each dataset, we find that the only non-trivial certification comes from training with our method. Even for PneumoniaMNIST where the standard network shows impressive attack robustness, we are unable to certify explanation robustness for standard trained models. This result mirrors the fact that only certified training yields non-vacuous prediction robustness bounds for high-dimensional inputs in the adversarial attack literature (Gowal et al., 2018). We highlight that our method indicates that no model attack can be successful if it perturbs the weights by 2.5% of their value, but we cannot make guarantees outside of that range, and in principle there will always exist a classifier outside of the guaranteed range that corrupts the input gradient. In the left half of Figure 4, we plot example explanations from NNs trained with standard training as well as our training method. To avoid cherry-picking bias we use the first test image for each dataset. We find that the explanations provided by networks trained with our method are more sparse and seem to be better correlated with intuitively important features of the input image.

## 7 CONCLUSION

We present a computational framework for upper-bounding the largest change an adversary can make to the input gradient of a neural network given bounded modification of the input features or model parameters. Our framework allows us to present, to our knowledge, the first formal certification of explanation robustness for general neural networks. We empirically validate our findings over six datasets of varying complexity and find that our method is able to produce certificates of explanation robustness for each.

## ACKNOWLEDGMENTS

MW acknowledges support from Accenture. MW and AW acknowledge support from EPSRC grant EP/V056883/1. AW acknowledges support from a Turing AI Fellowship under grant EP/V025279/1, and the Leverhulme Trust via CFI.

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
