# OpenReview forum: "Robust Explanation Constraints for Neural Networks"
_ICLR.cc/2023/Conference — ICLR 2023 poster_

### Official Review · Reviewer_wHym · 2022-10-17

**Confidence:** 4
**Correctness:** 2
**Technical Novelty And Significance:** 2
**Empirical Novelty And Significance:** 2
**Recommendation:** 5

**Clarity, Quality, Novelty And Reproducibility:**

Clarity: The paper is easy to read, however as mentioned in the weaknesses section, the state of some contributions are unclear and some mathematical notations are undefined.

Quality: The proofs are currently missing, which makes it impossible to assess the correctness and applicability of Lemma 1. Experiments miss crucial baselines, thus the paper scores low on quality.

Novelty: The problem proposed in the paper, that of simultaneous gradient robustness wrt inputs and weights, is novel as far as I am aware. IBP bounds are known, so the method scores low on novelty of methods.

Reproducibility: The calculations presented in the paper seem simple enough to be reproduced independently, so this does not appear to be an issue.


**Strength And Weaknesses:**

**Strengths**:
+ The problem considered is an important one, and the proposed direction of having certifiable bounds on gradients robustness has not been explored much in literature, and thus is novel and
interesting from that perspective.

+ The proposed solution of presenting a common method that provides robustness both to input and weights perturbation via box constraints is novel as far as I am aware.

**Weaknesses**:

**Lemma 1: unclear novelty, missing proof, unclear notations, unclear slackness**

- The central result for this paper (Lemma 1) seems to be a previously known result, and the contribution of this paper is its usage for forward and backward propogating interval bounds in NNs. However, this point is not stated clearly in the paper, as the introduction claims that "We derive explicit bounds relying on interval bound propagation" suggesting that they indeed derive Lemma 1. Perhaps they mean that they present a computational framework to **compute** bounds rather than derive them? It would be helpful to more clearly state the contributions of this paper in this case.

- The paper states that "We provide more detailed exposition and proof-sketch in Appendix X", which I could not find in the supplementary material. Presumably these are proofs for Lemma 1 and their usage for deriving NN forward and backward propagation?

- (Minor) The statement of Lemma 1 includes the notation M^B = A | B^r |. Just to clarify, is this a matrix multiplication of A with element-wise absolute value of B^r? It would be helpful to clarify this notation.

- (Minor) The paper mentions that this Lemma may not provides the tightest bounds. It would be helpful to provide some discussion or intuition about how loose these bounds are, and under what conditions are they tight.

- (Minor) I could not find Lemma 1 in the PhD thesis linked, so it would help to provide more detailed pointers regarding chapter number, etc.

**(Minor) Robustness definition may not be practically relevant**

- Definitions 1 and 2 present defns for explanation robustness in terms of box constraints. However, the **direction** of an explanation vector is more important than its magnitude which reveals the relative importance of various input features. Thus according to the current definition, assume there exists explanation vectors whose directions remains constant within inputs set "T" or parameter set "M", but only their magnitude changes. Then such models have non-robust gradients according to the present definition, but that may not be of practical significance as far as explanations are concerned.

**Missing baselines: gradient regularization, local linearization and curvature regularization**

- A missing baseline is gradient norm regularization, which penalizes the magnitude of the gradient explanation. The proposed regularizer penalizes both gradient magnitude and directions (see previous point), so it is important to disentangle the effects of magnitude vs direction regularization.

- Another set of missing baselines are a local linearity regularizer (https://arxiv.org/abs/1907.02610), and curvature regularization (https://arxiv.org/abs/1811.09716) which explicitly penalizes model curvature, which in the limit forces models to be linear similar to the proposed method.

- An alternative to these approaches is to compare against an adversarial training baseline, as all these methods are intended to be proxies for the same.

**Missing evaluations on larger datasets and models**

- Evaluations presently seem to be conducted on relatively smaller datasets (those that are MNIST-scale and below) and small models (two hidden layer neural networks). Some evaluations on larger datasets such as CIFAR / SVHN and larger models like deep resnets would help assess the scalability of the approach.

**Summary Of The Paper:**

This paper proposes certification methods for assessing input-gradient robustness to both perturbations in inputs and weights, and this is achieved using interval bound propagation (IBP) methods. To this end, the paper proposes to use box-type constraints to measure input-gradient robustness (definitions 1 and 2). The main methodological contributions are the usage of Lemma 1 on neural network, which presents a way to compute bounds of products of matrices; and section 5.1, which presents a regularized loss function based on the certified loss.

Experiments show that the proposed certified loss function results in models that have improved gradient attack robustness across datasets, and sparser gradients with minimal degredation of test performance.


**Summary Of The Review:**

Overall, this paper misses some crucial details about its central result that impact its novelty, misses relevant baselines, and conducts only small scale experiments. As a result, I am currently leaning towards a reject.

---

> ### Author Response · Authors · 2022-11-09
> **Response to Reviewer wHym**
>
> We thank the reviewer for their careful reading of our work and for the critical evaluation of our theoretical contributions. We have considered their input and have uploaded a revised paper. We highlight, however, that the reviewer may have missed several important points that were already discussed in the original submission and subsequently penalized the work in their review. Below we respond point-by-point to the criticisms of Lemma 1, and then discuss possible misconceptions in the review and how we have sought to make the manuscript more clear.
>
> ### On Lemma 1
>
> We have changed our diction to state that we compute bounds on Definition 1 and 2 and thank the reviewer for suggesting this change.
>
> We added Appendix I which includes the proof for Lemma 1 and a discussion of where it is tight and loose. For further discussion of where this bound could be improved (via more sophisticated optimization techniques), we refer the reviewer to Appendix A of the original submission.
>
> The reviewer is correct about $\vert \cdot \vert$ and this note has been added to Lemma 1.
>
> ### Definition of Robustness
>
> We agree that the focus of the definition is on the magnitude of the gradient change. However, we highlight that, as discussed in Section 5, the similarity function $h$ flexibly models situations such as the one described by the reviewer. To make this more clear to future readers we add a computation for sound bounds on the cosine similarity measure in Appendix H of the updated manuscript and highlight in the main text that change in the magnitude is simply one potential measure.
>
> ### On Missing Baselines
>
> We believe that this is a point that is missed by the reviewer and hope they will reconsider penalizing our work on this basis. Adversarial training w.r.t. the gradient norm, curvature regularization, and adversarial training are all baselines that are used as comparison in the original submission. In particular, these baselines are used in Figures 2, 5, 6, and 7 and discussed in several places. To bolster this point we have added a line in the updated main text, a section in the Appendix (Appendix G) and added two more figures (Figure 11 and 12).
>
> ### On Small Datasets
>
> We agree with the reviewer that large scale ImageNet experiments would be interesting, however, such experiments are beyond the capabilities of current convex relaxation methods for NNs. We have enhanced discussion of this limitation in the original Appendix A.

---

> > ### Comment · Reviewer_wHym · 2022-11-15
> > **Clarification regarding baselines**
> >
> > Thanks for your response!
> >
> > I think the authors may have misunderstood my comment about baselines. First, to avoid more confusion, it would help writing down explicit expressions for the regularizers used in the baselines in the appendix (with exact hyper-parameters used in these cases), as the papers linked sometimes propose multiple regularizers.
> >
> > Second, by adversarial training, I mean standard adversarial training as in Madry et al., "Towards deep learning models resistant to adversarial attacks", 2018; and not the variant in Chen et al., 2019 which is for adversarial attacks against explanations. I ask for this because the Moosavi Dezfooli et al., 2019 (https://arxiv.org/abs/1811.09716) paper shows that adversarial training implicitly performs Hessian regularization, which is central to the goals of this paper.
> >
> > Third, by input-gradient norm regularization I mean, adding an explicit penalty term of the form $ || \nabla_x f(x) ||^2$ to the standard loss function, which unless I am mistaken I do not see in the paper. This is the classic Drucker and Lecun, "Improving generalization performance using double backpropagation", 1992; which is also considered in proposition 2 of Chen et al., 2019.
> >
> > Regarding curvature regularization, I realise that the baseline I recommended (Moosavi Dezfooli et al., 2019) and the one you use (Dombrowski et al., 2022) can be considered roughly equivalent although their performance can differ vastly in practice, so I will consider a Hessian regularization baseline as being already presented. Note that Moosavi Dezfooli's method approximates standard adversarial training (as in Madry et al., 2018), whereas Dombrowski's methods do not.
> >
> > The authors claim "Adversarial training w.r.t. the gradient norm, curvature regularization, and adversarial training are all baselines that are used as comparison in the original submission". Unless I am mistaken, I see only adversarial training on explanations (Chen et. al., 2019); and Hessian regularization (Dombrowski et al., 2022) in Figure 2. Is this correct?

---

> > > ### Author Response · Authors · 2022-11-18
> > > **Further Baselines Confirm Results**
> > >
> > > We thank the reviewer for their clarification and agree that the original manuscript should have been more clear in its baselines w.r.t. (Chen et. al. 2019). We have now included the objective function and hyper parameters for each of the five baselines we compare with (several have been added at the reviewers request, details below). _The findings of the paper remain as they were before._ Our method proposed for optimizing the certified lower bound during training greatly outperforms all other methods in terms of explanation robustness. The robustness offered by our method is significantly stronger than other methods, thus, in our comparison, we reduce the adversaries strength (by reducing the perturbation budget) in order to observe non-trivial certifications for other baselines. That is, testing with the original perturbation budget leads to all baselines showing a certified robustness of 0.
> > >
> > >
> > > We think that the additional experiments have improved the paper and hope to add these baselines to the main text. For now, the baselines remain in Appendix G. Further, the baseline in the main text was in fact more in the spirit of (Drucker and LeCun, 1992) and we have changed the wording and amended this citation. Each of these changes is reflected in the now-revised manuscript. Below we summarize the changes:
> > >
> > >
> > > We have now added a direct comparison with PGD training (Madry et al., 2018), in the original manuscript we only compared to (Gowal et al., 2018) for adversarial training. In addition, we have implemented and compared the IG-NORM and IG-SUM-NORM methods, but have dropped the “I” initial as we only consider the gradient and not the integrated gradient as in (Chen et. al. 2019).
> > >
> > >
> > > Again, we thank the reviewer and appreciate their valuable suggestions, and we hope they will take the time to consider the new versions of Figures 11 and 12.

---

> > ### Comment · Reviewer_wHym · 2022-11-25
> > **Response to Rebuttal**
> >
> > I have gone through the updated paper and the rebuttal, and the following issues remain.
> >
> > (Relatively Minor) **Proof of Lemma 1: Proof incomplete; Inconsistency in lemma statement in appendix and in main paper**:
> > I thank the authors for presenting a proof of Lemma 1 in the appendix, however
> >
> > - It seems the proof requires computing the length of the interval, upper bounding it using Cauchy Schwartz. However this is not mentioned explicitly. Specifically, the proof states "we observe that the width of the interval between
> > the maximum and minimum (regardless of which is which) is $2A^r_{i,t}|B^μ_{t,j} |$". However the authors can verify that the width of the interval is actually $| 2 A^r_{i,t}B^μ_{t,j} |$, which is presumably upper bounded to $2A^r_{i,t}|B^μ_{t,j} |$ using Cauchy Schwartz. This view also clarifies my earlier question regarding when the bounds are exact, and that seems to happen when the angle between the rows of A and B are close to zero (due to Cauchy Schwartz). *To summarize, the proof misses discussion of its most critical tool (Cauchy Schwartz) used to obtain the bounds, and I hope the authors add these discussion points in an update to their draft.*
> >
> > - Inconsistency in lemma statement: In the main paper, Lemma 1 seems to involve bounding A_ij B_ij, whereas the proofs are for (AB)_ij, and similarly for all the terms in the inequality, which is not a minor error. In general, the lemma statement in the main paper is sloppily written, with the relevant indices misspecified in several places. I hope the authors are able to fix this in the next update of the paper.
> >
> > **Changing similarity function requires changing solution structure**
> >
> > In their rebuttal, the authors claim "similarity function $h$ flexibly models situations such as the one described by the reviewer". While this is true, changing the similarity function in such a manner requires changing the solution strategy as well, and the box-constraint type methods used in this paper will no longer apply to an arbitrary similarity function h. I hope the authors discuss this shortcoming in an update of their paper.
> >
> > **Fig 11 / 12 do not show clear superiority of the proposed method**
> >
> > On the topic of baselines, the rebuttal claims that "The findings of the paper remain as they were before", and that "Our method proposed for optimizing the certified lower bound during training greatly outperforms all other methods in terms of explanation robustness". However from Fig 11/ 12, it seems that Hessian regularization / GSUMNORM / gradient l2 regularization, are all on par with the proposed method in terms of the empirical attack robustness metric. While it is true the proposed method does achieve the highest certified robustness, it is important to note that loose bounds such as the certified robustness metric introduced in this work are unsuitable for comparison between methods (due to the slackness of the bounds introduced by usage of elementwise box-type constraints) and thus may not be a useful proxy to actual empirical robustness.
> >
> > **On larger datasets**
> >
> > In my review, I mentioned that experiments on larger datasets such as CIFAR/SVHN are desirable, but the rebuttal states "We agree with the reviewer that large scale ImageNet experiments would be interesting, however, such experiments are beyond the capabilities of current convex relaxation methods for NNs". First, this is a confusing response as CIFAR / SVHN are not imagenet-scale datasets. Second, it is important to make running time estimates to demonstrate that such experiments are infeasible.
> >
> > Thus overall, while I appreciate the authors updating the paper to improve clarity, the experimental results are ultimately not strong enough to merit acceptance in my view.

---

> > > ### Author Response · Authors · 2022-11-28
> > > **Clarifying Some Misconceptions**
> > >
> > >
> > > We greatly appreciate the reviewer’s time and commitment to reading and improving our paper. Below we respond to the reviewer – in summary we believe that the issues raised reflect misconceptions with the exception of one typo which is easily updated. We appreciate all the points and will further clarify in the paper accordingly, thanks!
> > >
> > > ## On the Lemma Formulation
> > >
> > > The values $A^r_{i,j}$ and  $|B^μ_{t,j}|$ are real values with $A^r_{i,j}$ being strictly non-negative by definition. Thus discussion of bounding by Cauchy-Swartz is unnecessary in this context as $2A^r_{i,t}|B^μ_{t,j} | = | 2 A^r_{i,t}B^μ_{t,j} |$. We have a discussion of tightness in Appendix I.2. While further discussion of the proof and more analysis of tightness along the lines raised by the reviewer may be useful, we suggest that this discussion is secondary to the contributions of the paper, where the results we presented are sufficient for certification of explanation robustness.
> > >
> > > Thank you for pointing out that the indices in the lemma statement could have been more clear. We have updated the Lemma to have the same indexing as the proof to avoid confusion.
> > >
> > > ## On change of Similarity Function
> > >
> > > Here, we are sorry if it was not clear how the $h$ function is used in our guarantees. For example, the paper states (w.r.t. Targeted Attacks) that “it is sufficient to check that $\forall v’ \in E, h(v, v’) > \tau$”. Notice that the fact that $E$ is defined as box constraints is independent of $h$. Where $h$ is convex (often the case for similarity metrics used for explanations), an optimal $v’$ can be found by solving a convex optimization problem. In the paper, we show for two commonly used similarity metrics that we can solve for these vectors analytically. In the paper these are the only two metrics we have provided, however, we will add further discussion in Appendix H that makes it clear that our method can easily handle any convex similarity function $h$, and thank the reviewer for the suggestion.
> > >
> > > ## Superiority of Our Method
> > >
> > > To avoid confusion, we’d like to emphasize a central point of the paper and critical details of Figures 11 and 12. First, the goal of the methodology is not to estimate or improve robustness to attacks per se. We seek to formally verify that no successful attack on explanation exists. To this end, no previous work can provide these guarantees and no previous work allows one to train networks such that these guarantees are strong. By carefully formulating a SAT or MILP problem it may be possible to achieve tighter bounds, but this is outside of the scope of the current paper and is discussed in Appendix A.
> > >
> > > Figures 11 and 12 indeed show our method is superior: our method achieves the main goal of the paper, which is certified explanation robustness. We highlight that in Figures 11 and 12, we have reduced the adversaries’ attack strength in order to compare other methods - this is highlighted in the text as well as in our response. Increasing the adversarial strength (e.g., to 0.5 for MNIST) reduces all other methods to 0 attack robustness and 0 certified robustness while leaving our method at 1.0 and 1.0 for both metrics.
> > >
> > > ## On Larger Datasets
> > >
> > > Here, we respectfully suggest that the reviewer may have misunderstood our scalability discussion - to avoid this confusion we have added a more clear section on this in Appendix A. State-of-the-art bound propagation techniques do not scale well to such large datasets/networks for one of two reasons. Either (1) the method’s complexity prevents it from having a feasible running time (e.g., MILP or SAT solvers); or (2) the approximation introduced by convex relaxation makes the bounds vacuous for practical purposes. Our proposed method falls into the second category. We highlight, however, that addressing this shortcoming is outside the scope of the paper and that our contribution of a framework to certify explanations is still novel and of considerable interest.
> > >
> > > Nevertheless, for interest: we find that over the MNIST test, our method certifies robustness of each image in an average of 0.7ms on a MacBook Pro with M1 chip. For the CNNs on MedMNIST, each image takes approximately 1.3ms on the same machine. We find that the complexity of our method, equivalent to four forward passes through the networks, is a better measure of computational cost. This is noted in the first paragraph of Section 5.
> > >
> > > ## In Summary
> > >
> > > We appreciate the reviewer’s time and effort in reviewing our work, and hope that we have resolved all the outstanding issues. We believe the remaining reasons given for rejection were not focused on the main contribution of our work and may reflect possible minor misconceptions about details which we have addressed above, and will clarify in the paper. Again, we thank the reviewer and are happy to answer any further questions.

---

> > > > ### Comment · Reviewer_wHym · 2022-12-03
> > > > **Thank you for clarification**
> > > >
> > > > Thank you for clarification regarding the proof of the Lemma, the authors are correct that the main tool used to create the bounds is element-wise bounding rather than Cauchy Schwartz as I had earlier assumed. Thank you also for clarifying and updating the Lemma statement. I will increase my score as a result.

---

### Official Review · Reviewer_MkfV · 2022-10-21

**Confidence:** 3
**Correctness:** 3
**Technical Novelty And Significance:** 3
**Empirical Novelty And Significance:** 2
**Recommendation:** 8

**Clarity, Quality, Novelty And Reproducibility:**

Clarity- and quality-wise, I believe the paper is well-written. Also, to the best of my knowledge, the proposed idea is novel. The paper is augmented with a few appendices although I am not sure if the source code is provided (could not find a link in the paper) to reproduce the results described.


**Strength And Weaknesses:**

First of all, I should admit that I am no expert on heuristic explanation methods in machine learning so I might have overlooked some of the technicalities of the paper. However, I did my best to assess its general idea and its merits.

I will start by listing the pros of the paper. I believe the paper is nicely written. The discussion is concise but informative enough for an average reader (like me) to follow. It clearly poses the problem with the existing explanation approaches and proposes a solution to the problem. Although I have not checked the proofs of the theoretical results, they make perfect sense to me. I should also say that experimental results look convincing to me.

On the negative side and despite my previous comments on the clarity, the paper mixes up the concept of explanations with the properties of the models themselves, which I find somewhat confusing. This can be observed in Definitions 1 and 2, which seemingly discuss the properties of an explanation while, in fact, they describe the properties of a model.

Here I should also say that despite the claims made on explanation robustness, the authors fail to define what kind of explanations their methodology actually supports. Indeed, there are numerous post-hoc explanation approaches and the explanations they produce may vary a lot in terms of both syntax and semantics. It would be great if the authors could elaborate on this in their rebuttal.

Finally, the authors overlook a large body of work on formal explainable AI, where abductive and contrastive explanations computed by means of formally reasoning about the target model are guaranteed to be sound (correct), i.e. no robustness issues apply to them. Furthermore, there are probabilistic extensions of the above, which are not discussed in the paper either. I believe these lines of work should have been discussed in the related work. I would also appreciate it if the authors could comment on this in the rebuttal.

**Summary Of The Paper:**

This paper proposes a framework for assessing the explanation robustness of neural network (NN) models in the context of both classification and regression tasks. This proposed framework is claimed to solve a number of issues pertaining to the recent heuristic explanation approaches that suffer from a wide range of relevant adversarial attacks, i.e. "out-of-distribution" attacks, among many others. In particular, the paper develops a method to upper-bound the largest change an adversary can make in order to fool the corresponding explainer. The framework also allows one to train an NN model that is guaranteed to be certified explanation robust subject to a given parameter delta. Experimental results obtained for a few datasets demonstrate the efficiency of the proposed framework.

**Summary Of The Review:**

Although the paper has a few issues outlined above, I am inclined to believe it makes an interesting contribution, which can be deemed sufficient for publication at ICLR.

---

> ### Author Response · Authors · 2022-11-09
> **Response to Reviewer MkfV**
>
> We thank the reviewer for the time to read our paper and for their valuable suggestions to improve the clarity of our work. We have incorporated their feedback into the updated paper. Below, we summarize the steps taken to clarify the points brought up by the reviewer below:
>
> ### On Discussion of Non-Gradient Explanation Methods
>
> We thank the reviewer for the suggestion. We have added Appendix J which discusses some of these works as well as a sentence referencing this section from the main text.
>
> ### On the Notion of Explanation Robustness
>
> In the view of this work, which concerns gradient-based explanations, an explanation is tied to the underlying model. That is, the explanation is taken to be the gradient of the NN at the point, thus robustness is a property both of the explanation as well as the underlying model. For other forms of explanations, we agree that this definition may need to change accordingly, and non-gradient based literature is now discussed in Appendix J as suggested by the reviewer. In addition we note that our definition works well for any method that uses the gradient as a source of information (e.g., SmoothGrad, IntegratedGradients, Saliency Maps). This notion of robustness may extend to methods such as LIME, but this is left to future work.

---

> > ### Comment · Reviewer_MkfV · 2022-12-05
> > **Thank you**
> >
> > I thank the authors for the reply and increased my score.

---

### Official Review · Reviewer_MnSc · 2022-10-25

**Confidence:** 3
**Correctness:** 3
**Technical Novelty And Significance:** 2
**Empirical Novelty And Significance:** 2
**Recommendation:** 6

**Clarity, Quality, Novelty And Reproducibility:**

### Clarity
The motivation and procedure are clearly explained and the paper is well-written. I just find the loss thing confusing (see questions).
The text in Figure 2 is too small.

### Quality
The discussion of previous work is very thorough. The experiments are of high quality. The theoretical justification is not very thorough though. Proof and theoretical guarantees are missing.

### Novelty
The topic of explanation robustness is not new. The author's approach is very related to previous work. The empirical results show superior performance very clearly.

### Reproducability
The authors say that they will provide code, but it is not provided for the reviewers so I cannot assess the reproducibility at this point. Please provide an anonymous GitHub repository or send the code with the supplementary material next time to make assessment possible,

**Strength And Weaknesses:**

### Strengths

- important topic
- theoretical justification/motivation
- comparison to previous works
- quantitative and qualitative evaluation looks very promising
- the authors address robustness to input AND model parameter manipulation


### Weaknesses

- a more thorough theoretical analysis of differences to previous work would have been interesting
- no evaluation on high dimensional image data (ImageNet)
- the authors claim a bit too much for very little theoretical justification. After all they are still just solving an optimization problem, so there are not really guarantees.

### Question and Remarks
- I don't quite understand why in section 3 you define the gradient explanation as the gradient of the LOSS of the neural network output. It is usually just defined as the gradient wrt the winning class neuron of the network... The loss $\mathcal{L}$ is also not defined anywhere. Is that the training loss? Why?
- What is $J$ in the Backward Pass w.r.t. Bounds equations on page 6?
- Where is the proof sketch and where is Appendix X?
- Why can your certification procedure from section 4.1 not also be applied to standard, L2 Regr and Hessian Regr approaches?
- You talk a lot about guarantees and provable robustness. But the theory does not provide these guarantees and empirically you would have to test all possible perturbation configurations in your defined ranges which you probably didn't do.

### Typos/ minor remarks
- in 4: Now, we proceed define what it $\rightarrow$ Now, we proceed to define what it
- in 5: As desired, Lemma 1 allows us to jointy propagate $\rightarrow$ As desired, Lemma 1 allows us to jointly propagate
- choosing $\mathcal{L}$ for loss and loser bound seems a bit unfortunate.
- in 5.1: For input robustness, implies that for all points in $\rightarrow$ For input robustness, this implies that for all points in

**Summary Of The Paper:**

The authors propose a bound on how much a gradient based explanation can be manipulated (by input OR model parameter manipulation). They then include the bounds into the training regime to create certifiably robust networks. They compare their approach to previous attempts to make neural network explanations more robust on several data sets. The thorough quantitative and qualitative evaluation shows the advantages of the authors' proposed methods.

**Summary Of The Review:**

Overall I think this is a decent contribution. The authors could improve their work by providing a more thorough theoretical analysis. They might also consider lowering their claims and discussing the limitations of their approach critically. I'm happy to change my recommendation when my questions are answered to a satisfactory degree.

---

> ### Author Response · Authors · 2022-11-09
> **Response to Reviewer MnSc**
>
> We thank the reviewer for their careful read of our paper and for their suggestions that have led to an improved manuscript (already uploaded onto OpenReview). Below, we answer the reviewer’s questions point-by-point and hope that we have sufficiently clarified all points of confusion.
>
> ### On the Loss function
>
> We agree that in the original version of the paper this was unclear. We have made a significant modification to page 4 to clarify. We quote the passage here for convenience:  "Typically, one takes the input-gradient of the predicted class to be the explanation; however, one may want to generate the class-wise explanation for not just the predicted class, but other classes as well (Zhou et al., 2016; Selvaraju et al., 2017). In addition, the adversarial robustness literature often considers attacks with respect to surrogate loss functions as valid attack scenarios (Carlini & Wagner, 2017). By keeping L flexible in our formulation, we model each of these possibilities."
>
>  The notation J was erroneously left in the paper. It has been corrected everywhere to refer to $\mathcal{L}$, the loss function.
>
> ### On the Proof and Guarantees
>
> The full proof for the Lemma as well as theoretical analysis of its tightness is now given in Appendix I.
>
> We highlight that the guarantees of this method are sound and formal, meaning that our certificates of robustness hold for all possible perturbation configurations around a given input. One only needs to check the upper and lower bounds resulting from Section 5, and if none of the reachable explanations are adversarial, then no adversarial perturbation exists for the region around the input. If by "optimization" the reviewer means minimizing the proposed training loss, then they correct in saying that we cannot guarantee that networks trained with our method will yield strong test-set guarantees. We can only guarantee that if the proposed objective is minimized on the training set, then at least on the training dataset the neural network has strong provable guarantees of explanation robustness.
>
> ### On Empirical Comparisons
>
> Our method for bounding the change to an explanation can apply to networks trained with any loss function or regularization method. For example, see Figure 2 or Figures 11 and 12 of the updated Appendix. If the reviewer is referring to using our method in conjunction with other regularization methods, then this is also possible one would simply sum all of the desired regularization terms together.
>
> ### Analysis and Limitations
>
> For discussion of limitations we refer the reviewer to Appendix A. For theoretical discussion and proof of guarantees, we thank the reviewer for the suggestion and point them to the newly added Appendix I which not only adds the proof but also a discussion of the tightness of our bound.
>
> We again thank the reviewer and note that we would be glad to respond to any further questions or comments.

---

> > ### Comment · Reviewer_MnSc · 2022-11-18
> > **Thanks**
> >
> > Thanks for the additional clarifications. I will increase my score from 5 to 6.

---

### Official Review · Reviewer_tiM5 · 2022-10-28

**Confidence:** 3
**Correctness:** 4
**Technical Novelty And Significance:** 3
**Empirical Novelty And Significance:** 3
**Recommendation:** 8

**Clarity, Quality, Novelty And Reproducibility:**

- The paper is well written, clear and of good quality.
- Reproducibility is somewhat harder to ascertain, although the experiments are described in details, the code link is not available.

**Strength And Weaknesses:**

Strengths:
- Important and significant research problem. Robustness of of explanation is an important problem from interpretability as well as fairness point of views. As authors point out, most current explainers are not very robust.
- Theoretically grounded with provable certifications.
- Actionable result as the results can directly be used to improve NN training from robust explanations point of view.
- Thorough experiments on a variety of different datasets.

Weaknesses:
These are minor points but can help
- the $\tau$ notation is overloaded between being a similarity threshold and a dissimilarity threshold. I don't think that confusion is needed either. Authors can stick with $\tau$ as similarity threshold and simply flip the inequalities where it's currently being used as a dissimilarity threshold, this will also improve readability.

- A few citations can enhance the intro further in the discussion of explainer robustness e.g. Alvarez-Melis & Jaakkola (2018) "On the robustness of interpretability methods" is a good reference discussing why robustness is important in explainers and how to quantify it. Khan et al. (2022) "Analyzing the effects of classifier Lipschitzness on explainers" and Wang et al. (2022) " "Robust Models Are More Interpretable Because Attributions Look Normal" " discusses the connection between smoothness of classifiers and robustness of explanation. Zhou et al (2022) "From local explanations to model understanding" and Ju et al (2022) "Logic traps in evaluating attribution scores" provide some useful counterpoints in the discussion about the importance of robustness for explainers.

**Summary Of The Paper:**

This paper provides certification guarantees on robustness of gradient based explainers by providing upper bounds on the largest adversarial change that can be made to the explanation from these explainers given bounded manipulation of either the input features or model parameters. Additionally, these bounds are differentiable, which means they can be used during training of neural networks that the explainers intend to explain and the resulting explanations will be provably robust.

**Summary Of The Review:**

I believe this paper is worth appearing in ICLR as it enhances our understanding of explainer robustness, and provides actionable tools for certifiably achieving it at training time.

---

> ### Author Response · Authors · 2022-11-09
> **Response to Reviewer tiM5**
>
> We would like to thank the reviewer for their careful read and review of our paper and for the kind comments. We have included some of the citations noted by the reviewer in the updated manuscript. As the reviewer points out, we hope this work can be used in future to provide actionable insights on how to improve explanations.

---

### Author Response · Authors · 2022-11-09
**Response Summary**

In this comment we summarize the state of discussion. First, we would like to thank each of the reviewers again for their time and effort. We believe that the edits we have made to our paper have improved the clarity of our work. We provide an updated log of changes at the bottom of this comment.

Since the original reviews we have appropriately addressed the concerns of Reviewer MnSc who has updated their score from a 5 to a 6. Thus 3 of the 4 reviewers now have accept scores. Our response to Reviewer wHym seemed to partially resolve their concerns, but in a new response they have raised further questions which we believe to be, primarily, based on minor misconceptions. We have addressed all these further questions in a response and will update the paper and Appendix accordingly to improve clarity.

Updated Change Log:

* Formal proof of Lemma 1 has been added in Appendix I, and theoretical discussion of its tightness has also been added.
* Related works discussing non-gradient-based explanations has been added in Appendix J.
* A paragraph on the role and choice of loss function has been added to page 4.
* Formal bounds on the cosine similarity metric are now given in Appendix H.
* Discussion of the class of suitable $h$ functions for our method is now in Appendix H.
* Further baseline comparisons (with 5 methods) on each dataset are now given in Appendix G (Figures 11 and 12)
* Minor notational corrections have been made as suggested by reviewers

We would be glad to respond to any further questions or comments.

---

### Comment · Area_Chair_67uh · 2022-11-18
**Responses**

Dear Reviewers,

Do you have any comments/replies to author's responses - it would be great if you could respond to them. Have they changed your opinion on the paper?

Kind regards,
AC

---

### Decision · Program_Chairs · 2023-01-20

**Decision:**

Accept: poster

**Justification For Why Not Higher Score:**

The method should have experimental evaluation on larger datasets/networks.

**Justification For Why Not Lower Score:**

The paper is interesting and novel and correct.

**Metareview: Summary, Strengths And Weaknesses:**

This paper provides a novel method to bound adversarial changes to gradient based explanations and train neural with certified robustness guarantees. The method and paper are well executed. The primary drawback is that the method has only been tested on MNIST scale experiments and likely does not scale to large models. Despite this the paper presents a novel and interesting contribution and should be published.

**Note From Pc:**

if the above contains the word "oral" or "spotlight" please see: "oral" presentation means -> notable-top-5% and "spotlight" means -> notable-top-25%. As stated in our emails, we are disassociating presentation type from AC recommendations